# Corrosion Activity of Carbon Steel B450C and Stainless Steel SS430 Exposed to Extract Solution of a Supersulfated Cement

**DOI:** 10.3390/ma15248782

**Published:** 2022-12-08

**Authors:** David Bonfil, Lucien Veleva, Sebastian Feliu, José Iván Escalante-García

**Affiliations:** 1Center for Research and Advanced Study (CINVESTAV), Applied Physics Department, Campus Merida, Merida 97310, Yucatán, Mexico; 2National Center for Metallurgical Research (CENIM-CSIC), Surface Engineering Corrosion and Durability Department, 8040 Madrid, Spain; 3Center for Research and Advanced Study (CINVESTAV), Campus Saltillo, Ramos Arizpe 25900, Coahuila, Mexico

**Keywords:** carbon steel, stainless steel, pumice, supersulfated cement, Portland cement, cement extract solution, corrosion potential, pH, SEM-EDS, XRD, XPS, EIS

## Abstract

Carbon steel B450C and low-chromium stainless steel SS430 were exposed for 30 days to supersulfated “SS1” cement extract solution, considered as a “green” alternative for partial replacement of the Portland cement clinker. The initial pH of 12.38 dropped since the first day to 7.84, accompanied by a displacement to more negative values of the free corrosion potential (OCP) of the carbon steel up to ≈−480.74 mV, giving the formation of γ-FeOOH, α-FeOOH and Fe_2_O_3_, as suggested by XRD and XPS analysis. In the meantime, the OCP of the SS430 tended towards more positive values (+182.50 mV), although at lower pH, and XPS analysis revealed the presence of Cr(OH)_3_ and FeO as corrosion products, as well the crystals of CaCO_3_, NaCl and KCl. On both surfaces, a localized corrosion attack was observed in the vicinity of local cathodes (Cu, Mn-carbides, Cr-nitrides, among others), influenced by the presence of Cl^−^ ions in the “SS1” extract solution, originating from the pumice. Two equivalent circuits were proposed for the quantitative analysis of EIS Nyquist and Bode diagrams, whose data were correlated with the OCP values and pH change in time of the “SS1” extract solution. The thickness of the corrosion layer formed on the SS430 surface was ≈0.8 nm, while that on the B450C layer was ≈0.3 nm.

## 1. Introduction

Carbon steel and stainless steel are common reinforcements for concrete structures. During the hydration (curing process), ordinary Portland cement forms up to (≈20%) of portlandite (Ca(OH)_2_), which alkalinizes the concrete pore solution (pH ≈ 13) [1] and it has a large buffer capacity, acting as a diffusion barrier upon the arrival of metal corrosive species [2,3,4,5]. Under that alkaline environment, a thin film of oxide/hydroxide corrosion products is formed on the embedded steel surface, known as passive layer, which prevents the development of the corrosion process. The composition of the passive layer and its structure (crystalline or amorphous) remains controversial, despite the use of in situ and ex situ techniques [6]. On the other hand, because of the self-generation of the stainless-steel passive film (rich in a chromium oxide surface layer), although the alkalinity of concrete diminishes, stainless steel reinforcement is preferred to the traditional carbon (mild) steel [7,8,9].

The existing concerns related to the environmental impact of the production of Portland cement (PC), which accounts for about 7–8% of the global CO_2_ emission [10], requires the partial replacement of the PC clinker with supplementary cementitious materials (SCM) [11].

A number of alternative cements [12,13,14,15] have been considered to reduce the environmental impact of the PC production, among which the supersulfated cements (SSCs) have been proposed as “green” cements. Generally, the formulation of SSC consists typically of 80–85% granulated blast furnace slag (GBFS) as precursor and 10–15% sulphatic and alkaline activators [16]; the common activators are gypsum (CaSO_4_.2H_2_O), anhydrite (CaSO_4_), hemihydrate (CaSO_4_∙1/2 H_2_O), Portland cement, CaO, and Ca(OH)_2_ [17,18,19]. Moreover, pumice (PM), a widely available low-cost type of volcanic silicoaluminate material [20], has been recently proposed as a precursor for SSC because is prone to chemical activation by alkalis and sulfates [21]. The use of PM has been studied in many applications, such as up to 50% substitution of PC [22,23], aggregates for lightweight concretes [24,25], and alkali-activated cements [26].

The SSCs develop strength, due to the formation of calcium silicate hydrates (C-S-H) and ettringite (sulfoaluminate, Ca_4_(AlO_2_)_6_SO_4_) as the main hydration products [27]. Other noteworthy features are the good durability in sulphatic environments and chemically aggressive environments, low heat of hydration, and a good mechanical performance at low temperatures [28]. However, the low PC content in alternative cements and the consumption of portlandite by the pozzolanic materials during the hydration process [29] could reduce the pH of the concrete pore solution, which may be detrimental to the passivity of embedded reinforcing steels, triggering the corrosion state [30].

The study of corrosion behavior of steel embedded in concrete presents difficulties in experimental measurements, including the electrode and cell designs, the position of reference and auxiliary electrodes, the large potential drop IR in concrete and its compensation, the restriction of oxygen diffusion, and the development of macro-corrosion cells, among others. In order to avoid many of these, the metals (electrodes) have been immersed in model solutions that simulate the concrete pore solution environment, including saturated aqueous Ca(OH)_2_ solutions (pH 12–13) [31,32,33], aqueous KOH and NaOH solutions [34,35,36,37], and cement extract (CE) solutions, because of the variety of ions present (Ca^2+^, Na^+^, K^+^, OH^−^, and SO_4_^2−^) [38,39,40,41,42]. Thus, the model solutions allow obtaining comparative results and to control some parameters, which are difficult to accomplish in reinforced-concrete samples.

In this study, the water extract solution of the supersulfated cement [21], named as “SS1”, based on 52% pumice (SiO_2_/Al_2_O_3_), 34% hemihydrate (${\mathrm{CaSO}}_{4}\cdot {1/2\;H}_{2}O$) as a sulfatic activator, and 7% of Portland cement (PC) and CaO as alkaline activators, was used to simulate the environment at the steel–concrete–pore interface. The electrochemical behaviors of carbon steel B450C and low-chromium ferritic stainless steel SS430 were characterized after exposure for up to 30 days. The change in time of the OCP (free corrosion potential) and pH of the extract cement solution as well the surface characterization (SEM-EDS and XRD) were performed. The obtained preliminary results were compared with those during the exposure to Portland cement extract [41]. To our knowledge, no other research on this topic has been previously undertaken.

## 2. Materials and Methods

### 2.1. Steel Samples and Surface Characterization

Flat samples of carbon steel B450C (Pittini Group, Gemona del Friuli, Italy) and low-chromium ferritic stainless steel SS430 (Outokumpu, Espoo, Finland) were cut (0.8 cm^2^), abraded with SiC paper to 4000 grit (with ethanol as a lubricant), then sonicated for 10 min (Branson 1510, Branson Ultrasonics Co., Danbury, CT, USA) and dried at room temperature (294 K or 21 °C). The nominal composition (wt.%), according to the manufacturers, is present in Table 1.

The surfaces of reference samples and those of samples after corrosion immersion tests were characterized by scanning electron microscopy (SEM-EDS, XL–30 ESEM-JEOL JSM-7600F, JEOL Ltd., Tokyo, Japan). The corrosion products were analyzed with X-ray photoelectron spectroscopy (XPS, K-Alpha, Thermo Scientific, Waltham, MA, USA), after sputtering the surface with a scanning Ar-ion gun for 15 s. The XPS spectra were calibrated by setting the main line for the O1s signal of oxygen in oxides at 530.2 eV, according to the suggested procedures for transition metal oxides [43]. Grazing incidence X-Ray diffraction patterns (Siemens D-5000, Munich, Germany; 2θ, 34 kV/25 mA CuKα), were analyzed to reveal the corrosion product composition.

SEM-EDS analysis [41] indicated two characteristic zones on the ferritic stainless-steel surface of SS430: one of a high Cr (24.03%) content in the presence of C (3.04%) and N (3.27%), ascribed to chromium nitride and carbide phases, as also to (Cr, Fe)_7_C_3_ phase and Cr-nitride [44,45]; another zone revealed aggregates of high C (17.19%) and Si (28.86%) content, attributed to silicon carbide (SiC). The detected V (0.7%) may be considered as replacing the sites of Cr in the Cr-C-N crystal structure, forming precipitates of V_6_C_5_ and VN-nitride phases, which block and prevent the grain growing, as well as the increase in ductility, hardness, and strength of the ferritic steel [46].

On the carbon steel B450C surface [41], SEM images showed black dots, whose EDS analysis has indicated that they contain carbon (5.02%), Mn (1.31%) and a lower content of S (0.38%), considered as a part of the phases of MnS and Mn3C [46,47]. The detected Cu content (0.83%) has been attributed to the quality of the scrap [48], while the contents of Si (6.46%) and carbon (8.49%) in some zones were ascribed to the SiC phase. Mn and Si are always present in carbon steel, although not explicitly reported by the supplier.

### 2.2. Supersulfated Cement and Extract Solution

The cement, labeled as “SS1”, was prepared according to previous reports [21,29], which indicated 49% lower CO_2_ emission during the cement manufacture, as well as a compressive strength of ≈44 MPa at 28 days (or better at 180 days), compared to that of PC-based concrete.

Table 2 presents the composition of SS1 according to the referred reports. The precursor for SS1 was volcanic pumice (51.72%), from the Perote region of Mexico, which is prone to chemical activation by alkalis and sulfates. The sulphatic activator is a hemihydrate CaSO_4_·1/2 H_2_O (34.48%), while the alkaline activators are Portland cement CPC30 (6.89%) and CaO (6.89%) [21]. The oxide composition is shown in Table 3.

According to reports [41,50], the Portland cement type I has 66.84–58.4% CaO as the main oxide, which is 2.5 times higher than in SS1 cement, followed by SiO_2_ (21.35–22.30%), Al_2_O_3_ (≈4.7%), Fe_2_O_3_ (2.89%), SO_3_ (2.42%), and MgO (1.16%), and at a low content are K_2_O (0.39–0.35%) and Na_2_O (0.08–0.28%).

In this study, the aqueous extract of the SS1 supersulfated cement is proposed as a model solution to simulate the non-carbonated concrete–pore environment. It was prepared from a 1:1 wt./wt. mixture of SS1 cement and ultrapure deionized water (18.2 MΩ cm). The mixture was agitated and left for 24 h to hydrate in a closed container. Then, the supernatant was filtered (2.5 µm pore size filter paper, Whatman, Kent, UK) to remove particles and kept in a sealed container. Table 4 presents the chemical composition of the SS1 cement extract solution, characterized by absorption spectrometry and atomic emission by plasma. The ion-selective technique was used to determine the Cl^−^ ion content.

### 2.3. Immersion Test

Triplicated steel samples (0.8 cm^2^) were immersed in 10 mL of SS1 cement extract solution for a period of 720 h (30 days), in sealed containers (with paraffin tape), according to the standard ASTM-NACE/ ASTM G31-12a [51]. After 168 h (7 days) and 720 h (30 days), the samples were withdrawn, rinsed with deionized water, and dried in air at room temperature (294 K or 21 °C). The corrosion layers were removed [52] and the surfaces were characterized by SEM-EDS and XPS. The composition of the corrosion layers was acquired by XRD analysis.

The pH of the cement extract solution was measured (PH60 Premium Line, pH tester. Apera Instruments, LLC, Columbus, OH, USA) after the steel samples were withdrawn at each period of exposure.

### 2.4. Electrochemical Measurements

A typical three-electrode cell configuration (inside a Faraday cage), connected to a potentiostat (Interface-1000E potentiostat/galvanostat/ZRA, Gamry Instruments, Philadelphia, PA, USA), was used for electrochemical experiments (294 K or 21 °C): the steel plates as working electrodes, the Pt plate as an auxiliary, and a saturated calomel electrode (SCE) as a reference electrode. The change in time of the open circuit potential (OCP), considered as the free corrosion potential of the studied steels, was monitored during the electrochemical experiments. Electrochemical impedance spectroscopy (EIS) at open circuit potential (OCP) was performed, applying an alternating current (AC) signal of ±10 mV amplitude, in a frequency range from 100 kHz to 10 mHz, and with a sampling size of 10 data points/decade. Nyquist and Bode EIS diagrams were recorded at different immersion periods: 24, 168, 360 and 720 h (30 days). The data were analyzed with Gamry Echem Analyst^®^ (version 7.1, Philadelphia, PA, USA).

## 3. Results

### 3.1. Change in Time of pH of the SS1 Cement Extract Solution

Table 5 presents the pH change in time of the SS1 cement extract solution during the exposure up to 30 days (720 h) of SS430 and carbon steel B450C. The initial pH of the SS1 cement extract solution was 12.38, while that of PC extract was 13.10 [41].

Since the first 24 h, the pH of SS1 cement extract tended towards a lower alkaline value of ≈9.58, and at the end of the experiment (30 days), the pH maintained a value of ≈7.82 (Table 5). This fact indicated that the carbon steel will lose its passive state because, since the first day of immersion, the pH value was below ≈11.5 [1]. (When the surface is covered with a thin oxyhydroxide film, known as a “passive film”, the steel is under a passive state.) The drop in the pH is mainly attributed to the consumption of OH^−^ ions for the formation of iron hydroxides/oxyhydroxides corrosion products. According to our previous study [41], during the exposure of the same steels to Portland cement extract solution, the pH of this solution shifted abruptly to a lower alkaline value of ≈9.10 at 14 days, and at the end of 30 days, the pH was ≈8.85.

### 3.2. Change in Time of the Open Circuit Potential (OCP)

The change in time of the OCP, considered as a free corrosion potential, is present in Table 6. The initial value of the carbon steel B450C (−206.54 mV) was more negative in ≈180 mV than that of SS430 (−20.28 mV). Since the first 24 h, the OCP of B450C shifted abruptly to a negative value of ≈−480 mV, while that of the SS430 tended towards a positive one (≈+104 mV), reaching at 30 days ≈+180 mV. These facts revealed that the SS430 preserved its passive state, whereas the B450C was in an active state of corrosion since the first 24 h. It is considered that when the OCP negative value is lower than −109 mV, there is a 90% probability of the corrosion process [53]. Compared to the exposure to the PC extract solution [41], the OCP of B450C was shifted to positive values up to 7 days (≈+113 mV), keeping its passive state, and then returned abruptly to extremely negative values of ≈−456 mV (an active corrosion state). In the meantime, the OCP of SS430 exposed to PC extract solution showed a firm tendency towards positive values, reaching at 30 days ≈217 mV [41], which is ≈30 mV more positive than that in the SS1 cement extract solution. Thus, it may be concluded that the SS430 presented similar passive states during its exposure to SS1 and PC extract solutions, not so influenced by the pH change in time.

The change in time of OCP values of the steels may be considered mainly because of the pH change in time (Table 5), which depends on the cement composition. According to reports [41,50], the Portland cement type I has 66.84–58.4% CaO as the main oxide, which is 2.5 times higher than that of SS1 cement (Table 3). Moreover, the supersulfated SS1 cement solution (Table 4) contains SO_4_^2−^ ions (563.5 mg/L), which are not present in the PC extract solution (or are below the limit of detection) [41]. Moreover, there are 45.4 mg/L of chloride ions, as a part of the SS1 cement extract solution (Table 4). Reported results [54] have suggested that in chloride–sulphate solutions (pH = 8), the mild steel did not passivate, and the corrosion resistance significantly decreased; however, the corrosion rate of the stainless steel AISI 315L was twice as low, presenting pitting corrosion influenced by the chloride ions, although this steel was in a passive state. Mass loss and electrochemical tests [55] revealed that sulphate ions could cause the corrosion of reinforcing steel in alkaline media (pH 9–12) and in the presence of chloride ions the attack is more severe, with evidence of pitting corrosion.

The shift in the OCP free corrosion potential to very negative values (−481.17 mV) after 7 days of exposure of the carbon steel to the SS1 extract cement solution (Table 6) confirmed the pH alkalinity importance needed for the steel surface passivation and, moreover, suggested the influence of the SO_4_^2−^ ions (Table 4) on the carbon steel corrosion (depassivation). Additionally, the chloride ions (45.4 mg/L, Table 4) may provide localized corrosion on the B450C surface.

### 3.3. Carbon Steel B450C Surface Characterization after Exposure to SS1 Cement Extract Solution

Figure 1 presents SEM images of the carbon steel B450C surface after the exposure for 7 and 30 days to the SS1 cement extract solution, and Table 7 reports the EDS analysis of several zones of interest. After 7 days, the formed corrosion layer presented several morphologies (Figure 1a), for which EDS analysis revealed that the content of Fe and O may be different (Zones A and B). In zone C, significant contents of Ca and carbon were observed, suggesting the formation of the CaCO_3_ crystals.

However, after the longer period of exposure of 30 days (Figure 1b,c), it seems that most of the CaCO_3_ crystals detached from the carbon steel surface because their mass and Ca content was not provided by the EDS analysis (Table 7, zone D). The cracking of the formed layer was observed at 30 days (Figure 1b,c) and the morphology of the corrosion layer was changed: it looked denser and EDS analysis indicated the presence of traces of Na (as a part of SS1 cement extract ionic composition, Table 4).

The chemistry of Fe-corrosion products is very complex and widely depends on the ion composition of the electrolyte in contact at the metal surface, as also influenced by the nature of the metal surface (in the presence of different chemical elements). Some ions, such as sulfates and chlorides, could promote secondary reactions and intermediate corrosion product formation: iron sulphate and chloride, with a short lifetime. The morphology of iron corrosion products changes over the time, attributed to different phases: sandy crystals, crystalline globules or fine plates have been reported for γ-FeOOH (lepidocrocite) [56]. The α-FeOOH goethite grows in a globular structure (known as cotton balls) [56,57,58,59,60,61,62,63]. Lepidocrocite and goethite have been reported as the main phases independently of the environmental composition [56,64,65].

X-ray diffraction spectra were used to reveal the composition of corrosion products formed on the B450C carbon steel surface, exposed for 30 days to SS1 cement extract solution (Figure 2). The spectra showed the characteristic peaks of γ-FeOOH (lepidocrocite), α-FeOOH (goethite) and Fe_2_O_3_ (hematite). Lepidocrocite is considered the final by-product of carbon steel corrosion compounds, while goethite is a transition phase. In aggressive environments (in the presence of Cl^−^), the lepidocrocite dehydrates and it is gradually transformed into different products, such as goethite and hematite [62]; the restriction of the diffusion of oxygen to the iron surface, due to the thicker layer of oxide/hydroxide corrosion products, leads to the reduction of such corrosion products [66].

### 3.4. Stainless Steel SS430 Surface Characterization after Exposure to SS1 Cement Extract Solution

Figure 3 shows the SEM images of stainless steel SS430 after the exposure for 7 and 30 days to SS1 cement extract solution and Table 8 presents the EDS element analysis of several zones of interest at 7 and 30 days of exposure.

As was indicated by the positive values of OCP (Table 6), the SS430 was in a passive state and the SEM images did not reveal a dense layer of corrosion products on the surface after 7 days of exposure to the SS1 extract solution (Figure 3a,b). EDS analysis (Table 8) suggested a high content of O, Na, K and S (in zone B), ascribed to the formed Na- and K-sulfates, deposited on the sites of the initial localized corrosion (influenced by the presence of Cl-ions), and thus it may consider that these sulfate deposits are attributed to the “repassivation” of those SS430 surface sites.

At 30 days (Figure 3c,d), several crystals were formed on the SS430 surface, whose composition was suggested by the EDS analysis (Table 8) as: CaCO_3_ and KCl in the zone Q1 and KCl in the zones Q2–Q3. The zone Q4 is characteristic for the SS430 matrix, in the presence of Na and Cl traces.

### 3.5. X-ray Photoelectron Spectroscopy (XPS) Spectra

Figure 4 presents the XPS spectra for SS430 and B450C steels exposed to SS1 cement extract solution for 30 days. The XPS spectra indicated that the main peaks detected for B450C were Fe and O (Figure 4b,d), while for stainless steel SS430, Cr was also detected (in addition to Na, Cl and K) (Figure 4c,g,h,i). The deconvolution of Cr2P was associated with Cr(OH)_3_ (578.08 eV), Cr_2_O_3_ (576.28 eV) and Cr metal (774.18 eV) [67]. The peak of Fe (710.6–711.6 eV) was associated with Fe^3+^ (Fe_2_O_3_/FeOOH) formed on the B450C and the binding energy of 709.5–709.7 eV was associated with FeO and Fe-metal (706.88 eV) on the SS430 surface [68]. The deconvolution of the peak of O1s showed the characteristic peaks of oxides (529.6–529.9), carbonates (530.78 eV) and OH^−^ (532.08 eV). The peaks of Ca2p, K2p, and Cl2p were associated with the formation of crystals of CaCO_3_ and KCl, suggested also by EDS analysis (Table 8), in more significant content on the SS430 surface [69].

### 3.6. Steel Surface Damage after Exposure to SS1 Cement Extract Solution

At 30 days of exposure to the SS1 cement extract solution, the formed layer on the stainless-steel surface of SS430 was removed (Figure 5). According to EDS analysis (Table 9), the Fe and Cr are the main elements of the metal matrix (zones A). The high content of Cr, C and N and the low level of V (zones Q2–Q3) were attributed to the Cr-C-N crystal structure and to vanadium as V (C, N) precipitates [46]. The Mn in zone B would act as cathodic active sites (reducing the oxygen), causing the dissolution of metal in the surroundings. The SEM images (Figure 5) showed corrosion pits on the SS430 surface, whose diameter was ≈1–3 µm.

After the exposure for 30 days to the SS1 cement extract solution, the formed layer on the B450C carbon steel surface was removed (Figure 6) and the EDS analysis is presented in Table 10. The existence of Cu (zone Q1) may be attributed to the quality of the scrap used to produce carbon steel, while zone Q2 corresponds to the matrix of the stainless steel. On the surface of carbon steel B450C, localized pits were observed, whose diameter was up to 7 µm.

### 3.7. Electrochemical Impedance Spectroscopy (EIS)

Figure 7 compares the Nyquist impedance diagrams of SS430 stainless steel and B450C carbon steel samples, exposed up to 30 days to SS1 extract solution. During the exposure (Figure 8a), the SS430 showed that at the low-frequency domain (10–100 mHz), the semi-linear diffusion impedance increased, associated with the diffusion control on the corrosion process, because of the formed passive film on the SS430 steel surface, mainly due to the Cr_2_O_3_ [67,70]. This fact collaborates well with the shift in OCP to positive values (Table 6), although the pH of the SS1 extract solution moved to less alkaline values. On the other hand, the exposed B450C carbon steel to SS1 extract solution (Figure 8b) showed a tendency towards a semi-circle at 1 day and not well-defined at 7 days, whose frequencies at the maximum point of the semi-circles are ≈126 mHz and ≈20 mHz, respectively. With increasing exposure time, the B450C displayed semi-linear diffusion Nyquist diagrams, whose impedance at low frequencies of 100 mHz showed a tendency of magnification in the time, although the pH of the SS1 extract solution shifted to lower alkalinity.

This fact does not corroborate with the more negative values of OCP (free corrosion potential), and it was attributed to the physical barrier on the carbon steel surface, provided by the formed corrosion product layer, giving a resistance to the steel corrosion process. However, because of the maintained passive state of the SS430 surface, the corresponding impedance value at 30 days (Z″ of 588 kΩ cm^2^) at low frequencies of 10 mHz was ≈17 times higher than that of B450C (Z″ of 35 kΩ cm^2^), presenting an active state of corrosion. This difference is also presented by the modulus of impedance as a part of the Bode diagrams (Figure 8a–c).

On the other hand, the Bode diagrams (Figure 8b–d) showed that the phase angle of the SS430 steel kept a value of ≈−80° (Figure 8b), closer to a phase angle of −90°, which indicates that the electrode interface was able to accumulate electrical charges and to block the migration of aggressive species (such as oxygen and chloride ions), because of the formed innert passive layer of low conductivity (mainly of Cr_2_O_3_). In the meantime, the phase angle of B450C carbon steel (Figure 9b) stabilized at −55° after 21 days of exposure. The lower phase angle confirmed the loss over time of the passive state as a consequence of the lesser alkaline pH of the SS1 cement extract solution.

In order to quantify the EIS data, two equivalent electric circuits (EC) were proposed (Figure 9), commonly used for the characterization of the electrochemical performance of carbon and stainless steels exposed to simulated concrete pore solution [71,72,73,74,75]. The simplified Randles circuit (Figure 9a) describes the electrochemical reactions of metals that develop a passive state (with only one time constant), where: Rs is the solution resistance at the electrode/electrolyte interface (which may change with the pH and ionic composition); Rct is the charge transfer resistance; and CPE_2_ is a constant-phase element, which replaced the double-layer capacitance of the interface (to obtain a better fit of the data) [72,75]. The second EC (Figure 9b), with two capacitances (time constants), was used to simulate the corrosion process of B450C carbon steel: the capacitance CP1 (high-frequency time constant) and Rcp (resistance to cracking propagation) are associated with the defective layer of (Fe_2_O_3_/FeOOH) formed on the B450C surface, while the CP2 (low-frequency time constant) is related to the double capacitive layer in the corrosion-localized area (film pores and active pits in the presence of aggressive ions) [75,76]. The exponential factor of CPE ranges from 0 to 1 (for an ideal capacitor *n* = 1) and *n* = 0 for an ideal resistor [68,77,78].

Table 11 presents the values of the fitting parameters obtained from the EIS measurements, and their fit c^2^ (10^−4^) was good in most cases. The polarization resistance (Rp) of steel, used as an indicator of the stability of passive films [79], is calculated by the sum of Rcp and Rct [67]:(1)$$R_{p}{=R}_{\mathrm{cp}}{+R}_{\mathrm{ct}}$$

Figure 10 compares the evolution of Rp values (Figure 10a,b) and passive-layer thickness (Figure 10c,d) during the immersion of SS430 and carbon steel B450C to SS1 supersulfated cement extract solution (this study) and to PC (Portland cement extract, in the absence of chlorides) [41]. It may be noted that at 30 days of exposure, the Rp value of SS430 in the PC extract solution was ≈3 orders higher than that in the SS1 extract solution (Figure 10a). However, the Rp values of B450C at that time were very similar: ≈107 kΩ cm^2^ in the SS1 cement extract and ≈56 kΩ cm^2^ in the PC extract solution (Figure 10b).

For the calculation of the *d* thickness, the CPE2 values were used, and transformed into the corresponding capacitance values, according to the Brug formula (Equation (2) [70]. The thickness was calculated from Equation (3) [80], where *ε_0_* is the vacuum permittivity (8.85 × 10^−14^ F cm^−1^) and ε is the dielectric constant of the passive film, which can be assumed as 15.6 for stainless steels [81,82].
(2)$${C=\mathrm{CPE}\;}^{\frac{1}{n}}{\left(\frac{R_{s}R_{\mathrm{ct}}}{R_{s}{+R}_{\mathrm{ct}}}\right)}^{\frac{1\;-\;n}{n}}$$
(3)$$d=\frac{{\varepsilon \varepsilon }_{0}A}{C}$$

It can be noted that during the exposure to the SS1 supersulfated cement extract, the thickness of the formed films on the SS430 surface was ≈0.8 nm (almost constant over the time), while that in Portland cement (PC) reached ≈1.8 nm at 30 days (Figure 10c). On the other hand, the thickness of the film formed on the B450C carbon steel surface, exposed to SS1 cement extract solution, increased over time, reaching ≈0.3 nm at 30 days, while the Portland cement extract (PC) attained a lesser thickness ≈0.15 nm, after the abrupt thickness diminishing at 15 days (Figure 10d).

## 4. Conclusions

The corrosion activities of low-chromium ferritic SS430 stainless steel and carbon steel B450C were studied during their exposure up to 30 days to SS1 supersulfated cement extract solution.

Initially, the pH of the SS1 extract solution (12.38) was lesser than that of the PC extract solution (≈13), mainly due to the minor quantity of CaO (26.07 wt.%) in SS1 cement composition than that in PC (≈62 wt.%). At the end of 30 days, the pH of the cement extract diminished, being 7.82 in the SS1 solution. In the meantime, the free corrosion potential (OCP) of B450C reached more negative values (−480 mV) and that of SS430 tended towards more positive values (+182 mV). These facts indicated that the B450C was in an active corrosion state, while the SS430 maintained its passive state.

At the end of the immersion test (30 days), on the B450C surface, the SEM-EDS and XRD analysis suggested the presence of γ-FeOOH (lepidocrocite), α-FeOOH (goethite) and Fe_2_O_3_ (hematite) as corrosion products, as well the crystals of CaCO_3_. The formed layer was cracked. On the SS430 surface, SEM-EDS and XPS analysis revealed the presence of Cr_2_O_3_; the corrosion products of Cr(OH)_3_ and FeO in the presence of well-formed crystals of NaCl and KCl (originating from the pumice); and, at lower content, the CaCO_3_ crystals.

On both steel surfaces, localized corrosion attacks were observed, influenced by the presence of Cl-ions (originating from the pumice). The EDS analysis suggested that the corrosion on the SS430 surface occurred in the vicinity of Mn (MnS), N and C, considered as local cathodes, while on the surface of B450C, it occurred mainly in the vicinity of Cu and Mn-carbides.

For the quantitative analysis of EIS (Nyquist and Bode diagrams), two equivalent electrical circuits (ECs) were proposed to characterize the corrosion activity of the studied steels at the metal–electrolyte interface. Due to the stable passive layer formed on the SS430 surface, the polarization resistance Rp kept a relatively constant value of ≈7870 kΩ cm^2^ (three orders lower than in the PC extract solution). It seems that the presence of SO_4_^2−^ ions (SS1 cement composition) led to the formation of a lesser protective and an unstable passive film on the stainless-steel surface, which could no longer be maintained, and local pitting corrosion occurred. On the other hand, the Rp values of B450C carbon steel were increasing in the SS1 supersulfated extract solution, being 1.6 times higher than that registered in the PC extract solution at 30 days of exposure. This fact was attributed to the slightly higher thickness of the formed corrosion layer on the B450C surface (≈0.3 nm) immersed in SS1 extract solution.

The reported results indicated that the change in time of pH and the OCP corrosion values are decisively dependent on the cement composition and that of the ions’ presence in the extract solution.

Supersulfated cement (SS1) may be recommended for concrete structures reinforced with carbon steel when they are exposed to the environment in the absence of aggressive chloride ions. In the presence of chlorides, the use of stainless steel in the concrete structure, based on SS1 cement, will depend mainly on the required service life for the structure.

## Figures and Tables

**Figure 1 materials-15-08782-f001:**
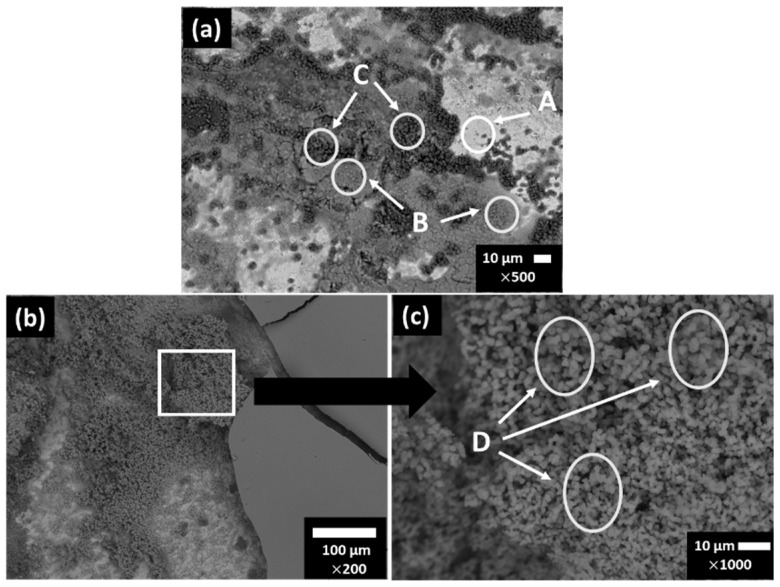
SEM images of carbon steel B450C exposed to SS1 cement extract solution after 7 days (**a**) and after 30 days (**b**,**c**).

**Figure 2 materials-15-08782-f002:**
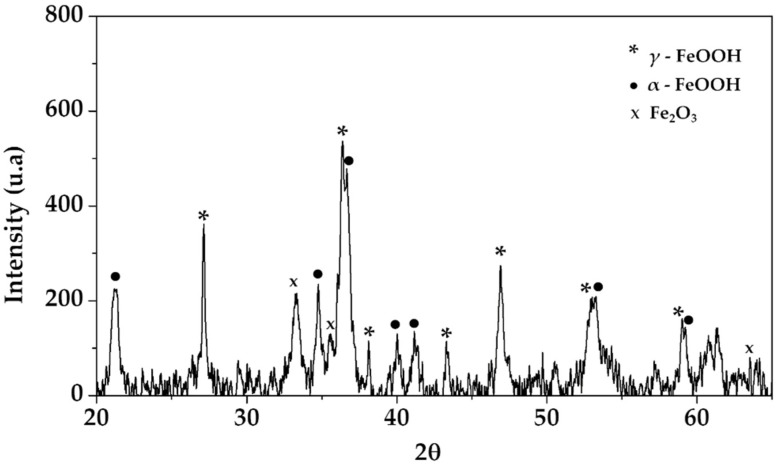
XRD spectra of the corrosion layer formed on carbon steel B450C surface, after exposure for 30 days to SS1 cement extract solution.

**Figure 3 materials-15-08782-f003:**
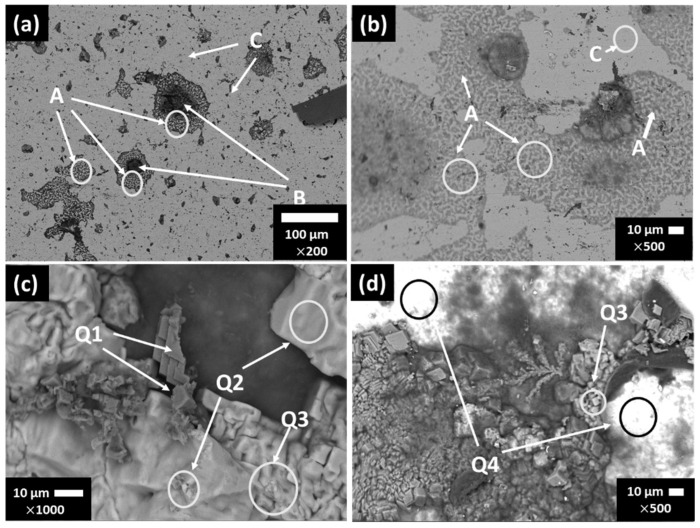
SEM images of stainless steel SS430 surface exposed to SS1 cement extract solution: (**a**,**b**) two different zones after 7 days and (**c**,**d**) two different zones after 30 days.

**Figure 4 materials-15-08782-f004:**
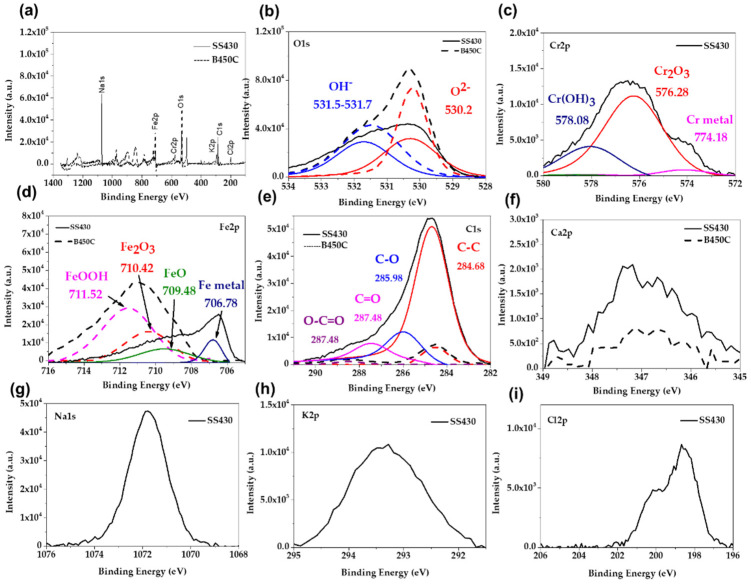
Overview of X-ray photoelectron spectroscopy (XPS) spectra acquired from SS430 and B450C steels exposed to SS1 cement extract for 30 days: (**a**) full spectrum; spectrum for (**b**) O1s; (**c**) Cr2p; (**d**) Fe2p; (**e**) C1s (**f**) Ca2p; (**g**) Na1s; (**h**) K2p; (**i**) Cl2p.

**Figure 5 materials-15-08782-f005:**
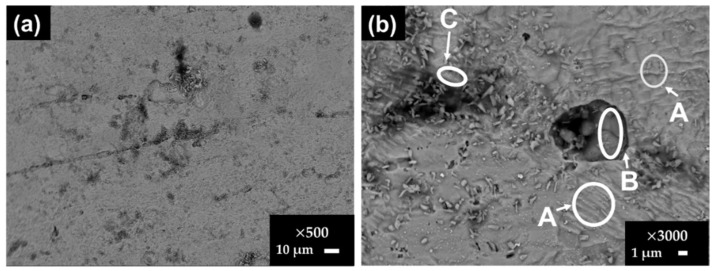
SEM images of stainless steel SS430 surface, after removal of the layer formed during the exposure for 30 days to SS1 cement extract solution: (**a**) ×500 and (**b**) ×3000 magnification (zones A, B, C).

**Figure 6 materials-15-08782-f006:**
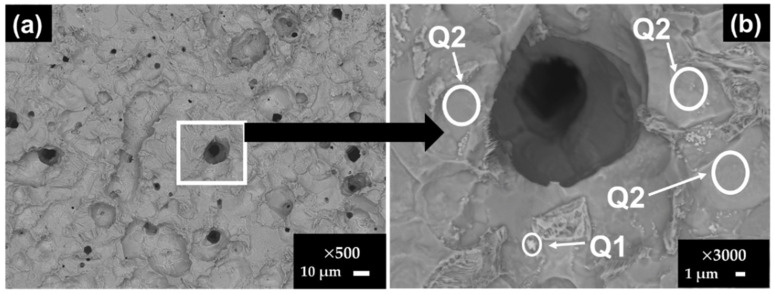
SEM images of carbon steel B450C surface, after removal of the layer formed during the exposure for 30 days to SS1 cement extract solution: (**a**) ×500 and (**b**) magnification ×3000 of the highlighted zone in (**a**).

**Figure 7 materials-15-08782-f007:**
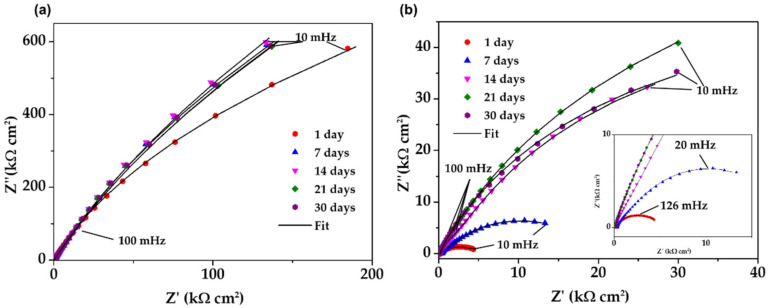
EIS Nyquist diagrams with a respective fitting line for SS430 and carbon steel B450C after different times of immersion in cement extract solutions: (**a**) SS430 and (**b**) B450C in SS1 solution.

**Figure 8 materials-15-08782-f008:**
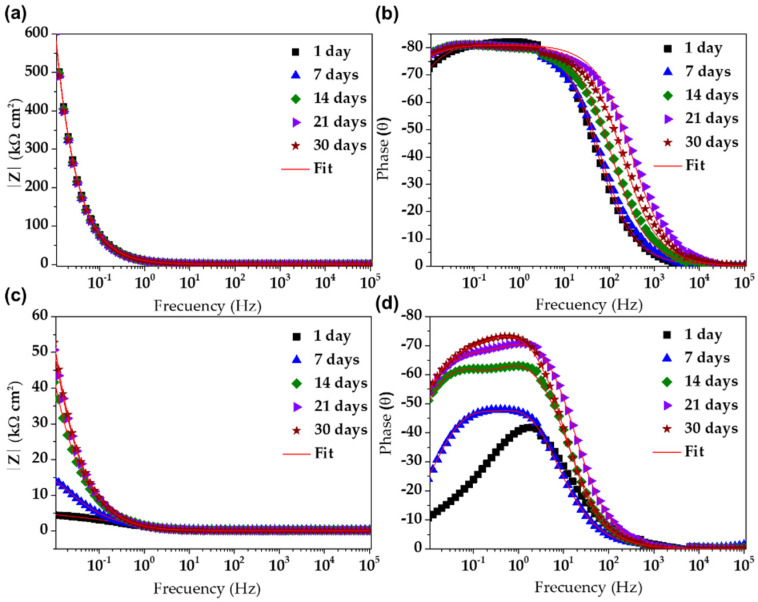
EIS Bode diagrams with a respective fitting line for SS430 (**a**,**b**) and carbon steel B450C (**c**,**d**) after different times of immersion in SS1 cement extract solution.

**Figure 9 materials-15-08782-f009:**
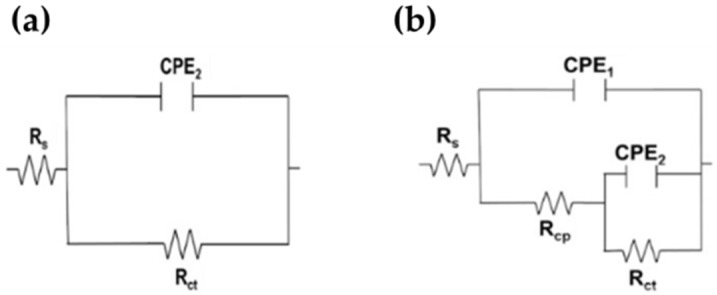
Equivalent circuits proposed for (**a**) SS430 steel and (**b**) carbon steel B450C exposed to SS1 cement extract solution up to 30 days.

**Figure 10 materials-15-08782-f010:**
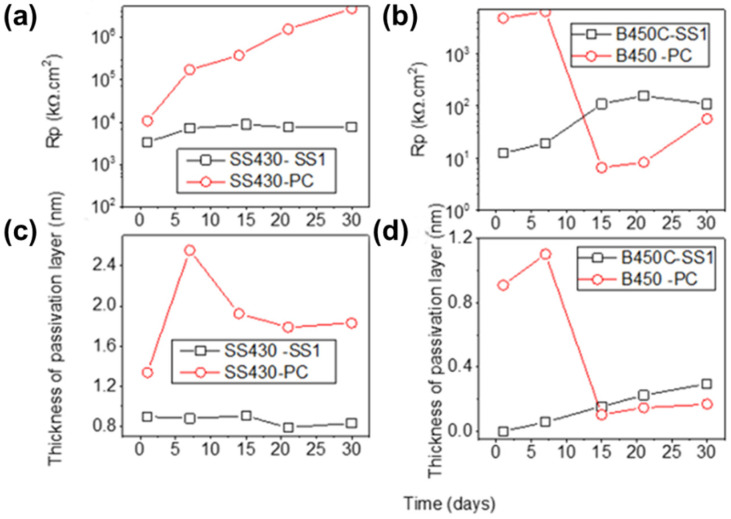
(**a**,**b**) Evolution of Rp values and (**c**,**d**) passive-layer thickness during the immersion of SS430 and carbon steel B450C to cement extract solutions of SS1 supersulfated cement (this study) and PC (Portland cement in the absence of chlorides) [41].

**Table 1 materials-15-08782-t001:** Compositions (wt.%) of SS430 ferritic steel and B450C carbon steel, according to manufacturers.

Element (wt.%)	C	Cr	N	Cu	P	S	Fe
SS430	0.25	16.2	-	-	-	-	Balance
B450C	0.22	-	0.12	0.8	0.5	0.5	Balance

**Table 2 materials-15-08782-t002:** Composition (wt.%) of SS1 supersulfated cement based on pumice, hemihydrated Ca-sulfate (Hh), cement CPC30 and CaO [21].

SS1	CPC30	CaO	Hh (CS) (CaSO_4_·1/2H_2_O)	Pumice (SiO_2_/Al_2_O_3_)	Total
wt.%	6.89	6.89	34.48	51.72	100

**Table 3 materials-15-08782-t003:** Oxide composition (wt.%) of SS1 supersulfated cement.

SS1	SiO_2_	Al_2_O_3_	Fe_2_O_3_	CaO	MgO	SO_3_	K_2_O	Na_2_O	TiO_2_	Cl	Others
wt.%	38.22	7.82	1.63	26.07	0.49	19.46	2.98	2.00	0.24	0.06	1.03

Others: MnO, P_2_O_5_, SrO, and ZrO_2_. The chlorides (Cl) are part of pumice [49].

**Table 4 materials-15-08782-t004:** Ion chemical composition (mg/L) of SS1 extract solution reported by Laboratory of Chemical Analysis (Cinvestav-Saltillo, Mexico) and Ion selective electrode (Cl-ion).

Element (mg/L)	Li^+^	K^+^	Na^+^	Al^3+^	Ca^2+^	Si	SO_4_^2−^	Sr	Cl^−^
SS1	0.184	496.8	310.2	0.185	396.6	5.4	563.5	9.02	45.4

**Table 5 materials-15-08782-t005:** Change in time of pH of the SS1 cement extract solution during the immersion of SS430 and carbon steel B450C up to 30 days.

pH vs. Time (Days)	Initial	1	7	14	21	30
B450C (SS1)	12.38	9.56	7.80	7.66	7.64	7.84
SS430 (SS1)	12.38	9.60	7.92	7.97	7.63	7.80

**Table 6 materials-15-08782-t006:** Change in time of the free corrosion potential values (OCP) during immersion of SS 430 and carbon steel B450C in SS1 cement extract solution up to 30 days.

OCP (mV) vs. SHE Time (Days)	Initial	1	7	14	21	30
B450C (SS1)	−206.54	−466.34	−481.17	−476.10	−477.44	−480.74
SS430 (SS1)	−20.28	104.17	171.51	173.56	178.92	182.50

**Table 7 materials-15-08782-t007:** EDS surface analysis (wt.%) of carbon steel B450C after exposure to SS1 cement extract solution for 7 and 30 days.

Days/wt.%		Fe	O	Mn	Zn	Ca	C	Na
7	A	85.02	9.27	0.89	-	-	4.81	-
	B	58.52	32.27	1.58	1.30	1.16	5.17	-
	C	26.18	44.43	0.82	-	**14.24**	14.14	-
30	D	48.85	47.55	0.48	-	-	1.81	0.96

**Table 8 materials-15-08782-t008:** EDS surface analysis (wt.%) of SS430 exposed to SS1 cement extract solution for 7 and 30 days.

Days/wt.%		Fe	O	Cr	C	S	Na	Cl	K	Si	Mn	Ca
7	A	58.67	5.90	12.86	15.04	0.52	2.67	0.50	0.28	0.78	-	-
	B	8.46	**40.20**	2.31	4.98	**14.81**	**14.60**	0.67	**13.02**	0.65	-	-
	C	77.84	1.26	16.59	2.61	-	-	-	-	0.73	0.98	-
30	Q1	1.80	**38.66**	0.53	**26.90**	-	1.99	**7.52**	**5.91**	0.53	-	**15.69**
	Q2	1.23	3.97	0.46	-	-	0.58	**44.98**	**48.78**	-	-	-
	Q3	0.93	11.98	-	28.89	-	0.68	**29.51**	**28.01**	-	-	-
	Q4	62.59	4.46	13.50	16.30	.	1.46	0.92	0.32	0.26	-	-

**Table 9 materials-15-08782-t009:** EDS surface analysis (wt.%) of SS430 after the removal of the corrosion layer formed during the exposure for 30 days to SS1 cement extract solution.

wt.%		C	Cr	Mn	O	V	N	S	Fe
SS430	A	2.35	16.92	0.77	0.83	-	-	-	78.83
	B	**16.37**	**27.69**	**9.78**	23.01	**1.13**	0.37	0.21	18.61
	C	**28.57**	**24.17**	-	8.86	**1.24**	**16.11**	0.28	20.07

**Table 10 materials-15-08782-t010:** EDS surface analysis (wt.%) of B450C after the removal of the corrosion layer formed during exposure for 30 days to SS1 cement extract solution.

wt.%		C	Mn	O	Cu	Fe
B450C	Q1	2.72	-	1.66	**64.08**	31.54
	Q2	0.84	0.54	-	-	98.62

**Table 11 materials-15-08782-t011:** Fitting parameters obtained from the EIS measurements for SS430 and carbon steel B450C exposed to supersulfated SS1 up to 30 days.

Steel	Days	R_sol_ kΩ cm^2^	R_cp_ kΩ cm^2^	CPE_1_ μSs^n^cm^−2^	n_1_	R_ct_ kΩ cm^2^	CPE_2_ μSs^n^cm^−2^	n_2_	R_p_ kΩ cm^2^	c^2^ 10^−4^
SS430	1	0.24	-	-	-	3356	19.74	0.91	3356	1.86
	7	0.21	-	-	-	7217	19.81	0.91	7217	0.1
	14	0.14	-	-	-	8839	19.06	0.9	8839	0.23
	21	0.05	-	-	-	7622	19.4	0.9	7622	0.29
	30	0.08	-	-	-	7806	19.41	0.9	7806	0.24
B450C	1	0.29	2.89	186.6	0.73	9.567	972.6	0.25	12.46	1.37
	7	0.36	3.98	160.1	0.81	14.94	222.3	0.75	18.92	0.98
	14	0.21	11.06	121.7	0.85	97.68	89.06	0.79	108.74	3.64
	21	0.12	8.97	116	0.9	145.7	63.69	0.71	154.67	0.44
	30	0.2	13.77	140.6	0.9	93.81	61.58	0.7	107.58	0.53

## Data Availability

The data are available upon request from the corresponding author.

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
