# Peer review of "Corrosion Activity of Carbon Steel B450C and Stainless Steel SS430 Exposed to Extract Solution of a Supersulfated Cement"

_materials, 2022, doi:10.3390/ma15248782_

Round 1

Reviewer 1 Report

Bonfil et al. studied the correlation activity of carbon steel b450c and stainless steel ss430 exposed to extract solution of a supersulfilled cement. The manuscript is innovative, but there are some problems in the following aspects that need to be revised:

1 The abbreviation that first appears in the manuscript should be given the full name.

2 The language of the manuscript needs to be greatly improved.

3 Why did the excitation signal of electrochemical impedance spectroscopy choose 10 mV amplitude?

4 The temperature in the manuscript is suggested to be changed to international unit K.

5 Electrochemical impedance spectroscopy should be analyzed in more detail, Author's reference: colloids and surfaces a: physical and engineering aspects 645 (2022) 128892 and journal of colloid and interface science 609 (2022) 838–851.

6 Nyquist diagrams should give some frequency values, and the x-axis and y-axis should be equal.

Author Response

Reviewer 1: Bonfil et al. studied the correlation activity of carbon steel b450c and stainless steel ss430 exposed to extract solution of a supersulfilled cement. The manuscript is innovative, but there are some problems in the following aspects that need to be revised:

  1. The abbreviation that first appears in the manuscript should be given the full name. Response: accepted recommendation.
  2. The language of the manuscript needs to be greatly improved. Response: it vas improved.
  3. Why did the excitation signal of electrochemical impedance spectroscopy choose ±10 mV amplitude? Response: A smaller amplitude of perturbation the steel surface is closer to its real state of not applied external polarization (open circuit potential)
  4. The temperature in the manuscript is suggested to be changed to international unit K. Response: accepted.
  5. Electrochemical impedance spectroscopy should be analyzed in more details. Response: given below the p.6.
  6. Nyquist diagrams should give some frequency values, and the x-axis and y-axis should be equal. Response: the Nyquist diagrams were analyzed at different frequency values and the discussion was in collaboration with the change in time of pH of the extract solution and that of OCP corrosion values. For the Bode diagrams the values of impedance modulus was incorporated at the y-axis and in a separated figure was presented the phase angel. The use of equal x-axis and y-axis was not possible.

The authors appreciate the comments and suggestion very much, which helped for the improving of the submitted article.

Reviewer 2 Report

The electrochemical behaviors and surface morphology of carbon steel B450C and low chromium ferritic stainless steel SS430, were characterized after exposure in the supersulfated cements for 30 days, and the obtained preliminary results were compared with those during the exposure to Portland cement extract. The experimental content in the manuscript is substantial, and various experimental phenomena are explained in detail, but there are also some deficiencies for the author's reference.

1. Line 204-205. Was the immersion test exposed to air during the whole process? If yes, then the carbonization of CO2 will also consume OH-, please explain. The description of the experimental phenomenon should be rigorous and reasonable.

2. Line 237-238. The effect of Cl- on the passivation film should not be ignored as 45.4mg/L has been mentioned above, please add a description.

3. Line 246-248. As can be seen in Table 6, the pH decreased to 9.56 after 1 day. In fact, ordinary carbon steel cannot remain passivated after pH less than 10, even without the influence of SO42-.Therefore, the description in the article is not rigorous.

4. Line 475-477. According to Table 6 and Table 7, the pH of SS1 is always lower than that of PC, and the OCP of B450C in SS1 is also always lower than that in PC. More importantly, from the comparison between Figure 7a and c, the local corrosion of B450C in SS1 also seems to be more serious than that in PC. However, after 15 days in Figure 11b, the Rp of B450C in SS1 is higher than that in PC. This result seems to contradict the previous results, please explain.

5. Line 505-508. This sentence seems to contradict line 496. May I ask whether Cl- or SO42- affects the pitting corrosion of passivated film specifically? It is suggested to add relevant discussion above.

6. The manuscript gives a good description of pH, OCP, corrosion morphology before and after rust removal and EIS results, but it seems that there is no discussion part to correlate the above results. It is suggested that the author should add a section to discuss the reasons why B450C and SS430 show different corrosion behaviors in the two kinds of concrete.

Author Response

Reviewer 2: The electrochemical behaviors and surface morphology of carbon steel B450C and low chromium ferritic stainless steel SS430, were characterized after exposure in the supersulfated cements for 30 days, and the obtained preliminary results were compared with those during the exposure to Portland cement extract. The experimental content in the manuscript is substantial, and various experimental phenomena are explained in detail, but there are also some deficiencies for the author's reference.

  1. Line 204-205. Was the immersion test exposed to air during the whole process? If yes, then the carbonization of CO2 will also consume OH-, please explain. The description of the experimental phenomenon should be rigorous and reasonable. Response: Triplicated steel samples (0.8 cm2) were immersed in 10 mL of SS1 cement extract solution for a period of 720 h (30 days), in sealed containers (with a paraffin tape), according to the standard ASTM-NACE/ ASTM G31-12a [51]. Thus, the carbonization of CO2 should be not considered, as also the consume of OH- ions.
  2. Line 237-238. The effect of Cl- on the passivation film should not be ignored as 45.4mg/L has been mentioned above, please add a description. Response: The Cl-ion effect is only considered for the initiation of the pitting corrosion when the pore size of the formed films allows the penetration and diffusion of chloride ions. There is a text in our article (lines 225-231): Reported results [54] suggested that in chloride-sulphate solutions (pH=8) the mild steel did not passivate, and the corrosion resistance significantly decreased, however, the corrosion rate of the stainless steel AISI 315L was twice lower, presenting pitting corrosion, influenced by the chloride ions, although this steel was in a passive state. Mass loss and electrochemical tests [55] revealed that sulphate ions could cause corrosion of reinforcing steel in alkaline media (pH 9-12) and in the presence of chloride ions the attack is more severe, with evidence of pitting corrosion.
  3. Line 246-248. As can be seen in Table 6, the pH decreased to 9.56 after 1 day. In fact, ordinary carbon steel cannot remain passivated after pH less than 10, even without the influence of SO42-Therefore, the description in the article is not rigorous. Response (lines 183-185): This fact indicated that the carbon steel will lose its passive state because since the first day of immersion the pH value was below of » 11.5 [1]. (Yes, the carbon steel cannot remain its passive state, even without the influence of SO4-ions).
  4. Line 475-477. According to Table 6 and Table 7, the pH of SS1 is always lower than that of PC, and the OCP of B450C in SS1 is also always lower than that in PC. More importantly, from the comparison between Figure 7a and c, the local corrosion of B450C in SS1 also seems to be more serious than that in PC. However, after 15 days in Figure 11b, the Rp of B450C in SS1 is higher than that in PC. This result seems to contradict the previous results, please explain. Response: We accepted the recommendation of the Reviewer No.3 and the Figures and Tables which present results of the studied steel in PC extract solution (previously reported [41]) were excluded (except of the final Figure 10). However, there are short resumes included to help the readers, which explain the mentioned doubts.
  5. Line 505-508. This sentence seems to contradict line 496. May I ask whether Cl-or SO42- affects the pitting corrosion of passivated film specifically? It is suggested to add relevant discussion above. Responses: Lines 467-468: On both steel surfaces localized corrosion attacks were observed, influenced by the presence of Cl-ions (originated from the pumice). Lines 473-478: For quantitative analysis of EIS (Nyquist and Bode diagrams), two equivalent electrical circuit (EC) were proposed to characterize the corrosion activity of the studied steels at the interface metal-electrolyte. Due to the stable passive layer formed on SS430 surface, the polarization resistance Rp keeps a value relatively constant of ≈7,870 kΩ cm2 (3 orders lower than in PC extract solution). It seems that the presence of SO42- ions (SS1 cement composition) leaded to the formation of a lesser protective and an unstable passive film on the stainless steel surface, which cannot longer be maintained, and local pitting corrosion occurred. (More details are also available in the discussion of the results).
  6. The manuscript gives a good description of pH, OCP, corrosion morphology before and after rust removal and EIS results, but it seems that there is no discussion part to correlate the above results. It is suggested that the author should add a section to discuss the reasons why B450C and SS430 show different corrosion behaviors in the two kinds of concrete. Response: please see our response at p.4.

The authors appreciate the comments and suggestion very much, which helped for the improving of the submitted article.

Reviewer 3 Report

Reviewer Recommendation and Comments for manuscript materials-2075209 with the title: “Corrosion Activity of Carbon Steel B450C and Stainless Steel SS430 Exposed to Extract Solution of a Supersulfated Cement”, authors: D. Bonfil, L. Veleva, S. Feliu Jr., J.I. Escalante Escalante-García.

The authors present the study of carbon steel B450C and stainless steel SS430 corrosion for 30 days in supersulfated cement extract solution.

The article may be published after revision.

The main comments that I find useful for improving the quality of the article are presented below:

*Authors' names need to be checked and corrected. There is a difference between the names in the article (José Iván Escalante García) and those on the MDPI website (José Iván Escalante Escalente-García).

*Figure 1 and Table 2 must be deleted. They are already published and the corresponding comments should be removed.

*Table 6. Change in time of pH SS430 and carbon steel B450C for 30 days, in Portland cement (PC) extract [42] must be deleted. They are already published and the corresponding comments should be removed.

*Table 7. OCP variation during immersion of SS430 and carbon steel B450C for 30 days in Portland cement (PC) extract [42] must be deleted. They are already published and the corresponding comments should be removed.

*Figure 6c must be deleted. This is already published and the corresponding comments should be removed.

*Figure 7c must be deleted. This is already published and the corresponding comments should be removed.

*Figure 8c and 8d must be deleted. They are already published and the corresponding comments should be removed.

*Table 13. Fitting parameters obtained from the EIS measurements for SS430 and carbon steel B450C 462 exposed to PC [42] cement extract solutions up to 30 days must be deleted. They are already published and the corresponding comments should be removed.

*Figura 11. The red curves (passive layer thickness variation over time) correspond to already published data and should be deleted, as well as the related comments.

*The typos must be corrected.

L24. extract..

L50,51,58, . SO4.2H2O. subscript must be used

L58. Sulfoalumunate

SS430 or SS 430

ml / mL

etc.

*The Materials journal require a specific format of references, authors must pay more attention in their writing. e.g. capitalization

*There are some grammar and typing mistakes.

*The authors must revise the entire manuscript.

Author Response

Reviewer 3: The authors present the study of carbon steel B450C and stainless steel SS430 corrosion for 30 days in supersulfated cement extract solution. The article may be published after revision.

The main comments that I find useful for improving the quality of the article are presented below:

 Authors' names need to be checked and corrected. There is a difference between the names in the article (José Iván Escalante García) and those on the MDPI website (José Iván Escalante Escalente-García). Response: The mentioned authors will keep his name as José Iván Escalante-García.

Figure 1 and Table 2 must be deleted. They are already published and the corresponding comments should be removed. Response: the have been deleted and well resumed texts were included to help the readers.

Table 6. Change in time of pH SS430 and carbon steel B450C for 30 days, in Portland cement (PC) extract [42] must be deleted. They are already published and the corresponding comments should be removed. Response: the Table 6 was excluded and well resumed text was included to help the readers.

Table 7. OCP variation during immersion of SS430 and carbon steel B450C for 30 days in Portland cement (PC) extract [42] must be deleted. They are already published and the corresponding comments should be removed. Response: Table 7 was removed and well resumed text was included to help the readers.

Figure 6c must be deleted. This is already published and the corresponding comments should be removed. Response: Figure 6c was removed and well resumed text was included to help the readers.

Figure 7c must be deleted. This is already published and the corresponding comments should be removed. Response: Figure 7c was removed and well resumed text was included to help the readers.

Figure 8c and 8d must be deleted. They are already published and the corresponding comments should be removed. Response: Figures 8c and 8d were removed and well resumed texts were  included to help the readers.

Table 13. Fitting parameters obtained from the EIS measurements for SS430 and carbon steel B450C 462 exposed to PC [42] cement extract solutions up to 30 days must be deleted. They are already published and the corresponding comments should be removed. Response: The fifing parameters of both steel exposed to PC extract solution were deleted and well resumed text was included to help the readers.

Figure 11. The red curves (passive layer thickness variation over time) correspond to already published data and should be deleted, as well as the related comments. Response: Figure 11 (now as Figure 10) includes the data presenting the behavior of both steels exposed to SS1 cement extract and PC extract solutions. They are necessary, because the fitting parameters for PC extract solution were excluded from the Table 12 (new number). However, they are necessary for the reader, helping to understand our final conclusions.

The typos must be corrected. Response: all mentioned typos were corrected.

The authors appreciate the comments and suggestion very much, which helped for the improving of the submitted article.

Reviewer 4 Report

The following points need to be addressed

1) Line 84 - How have the authors confirmed that no other research on this topic has been previously undertaken? 

2) Line 88 - Steel Samples. Provide an actual image of samples

3) Provide the reference of Table 1. 

4) In Figure 1. What is the difference between carbon steel B450C with (×500) and  (×3000)? 

5) Line 147 rewrite. I have?

6) Line 166 - Provide the impression test samples

7) Line 235 - What is below the limit of detection?

8) Line 242 - corrosion rate of AISI 315L was twice lower, since it corroded in the passive state. Provide appropriate reference

9) Avoid self-citation. 

Author Response

The following points need to be addressed

1) Line 84 - How have the authors confirmed that no other research on this topic has been previously undertaken? Response: There is not reported article, where the behaviour of B450C and SS430 have been exposed to extract of SS1 as a supersulfated cement based on pumice

2) Line 88 - Steel Samples. Provide an actual image of samples.

3) Provide the reference of Table 1. Response: The authors accepted the suggestion of Reviewer No.3, to eliminate the Figure 1. This as an actual SEM image of the reference samples.

4) In Figure 1. What is the difference between carbon steel B450C with (×500) and  (×3000)? Response: The x3000 is as magnification of the area marked in x500.

5) Line 147 rewrite. I have? Response: Portland cement type 1 has …..(line 137).

6) Line 166 - Provide the impression test samples. Response: the steel samples used in this study are presented in the lines 153-154.

7) Line 235 - What is below the limit of detection? Response: This information was used as cited by the authors of Ref.50.

8) Line 242 - corrosion rate of AISI 315L was twice lower, since it corroded in the passive state. Provide appropriate reference. Response: The Reference No.54 is the appropriate as references, however the AISI 316 L is the correct stainless steel. (Lines 225-229:

9) Avoid self-citation. Response: Several References (39, 40, 41, 56) are needed approving our discussion. They should not be considered as self-citation. Besides, the self-citations do not help the authors for their h-rating.

The authors appreciate the comments and suggestion very much, which helped for the improving of the submitted article.

Round 2

Reviewer 3 Report

Reviewer Recommendation and Comments for manuscript materials-2075209 with the title: “Corrosion Activity of Carbon Steel B450C and Stainless Steel SS430 Exposed to Extract Solution of a Supersulfated Cement”, authors: D. Bonfil, L. Veleva, S. Feliu Jr., J.I. Escalante-García.

The authors provide a revised form of the manuscript in which they include the reviewer's comments and their own considerations. I believe that this revised form is much improved, but the authors should still check and correct any errors before publication.

*Table 3. 6.89+6.89+34.48+51.72=99.98. 0.02 is missing!?

*Figure 8. EIS Bode diagrams. A new series has been introduced (figure 8d). Legend is missing. Please compare Figure 9b in the original manuscript and Figure 8d in the revised version.

Author Response

The authors provide a revised form of the manuscript in which they include the reviewer's comments and their own considerations. I believe that this revised form is much improved, but the authors should still check and correct any errors before publication.

*Table 3. 6.89+6.89+34.48+51.72=99.98. 0.02 is missing!?

Response: The values in the Table 3 give exactly the sum of 100 %.

*Figure 8. EIS Bode diagrams. A new series has been introduced (figure 8d). Legend is missing. Please compare Figure 9b in the original manuscript and Figure 8d in the revised version.

Response: The new legend is: Figure 8. EIS Bode diagrams with a respective fitting line for SS430 (a-b) and carbon steel B450C (c-d) after different times of immersion in SS1 cement extract solution.

The legend of the time (in days) was also included in the Figure 8 (a,b,c,d).

In the Figure 9b of the original (first version of the submitted manuscript) the data line of 1 day was missing. For this reason, the Figure 8d (of the revised version) the corresponding values of 1 day were include. Thus, comparing both figures there is a difference.

In the Figure 9b of the original (first version of the submitted manuscript) the data corresponding to 1 day were missing. For this reason, the Figure 8d (of the revised version) includes the values corresponding to 1 day.

Thanks for your suggestions.

Reviewer 4 Report

Now the manuscript is accepted for publication 

Author Response

Thanks for your time and revision of our article.